# Surface roughness, plaque accumulation, and cytotoxicity of provisional restorative materials fabricated by different methods

**Rashin Giti**[1]◉, **Shima Dabiri**[1]◉, **Mohammad Motamedifar**[2]‡, **Reza Derafshi**[1]◉*

**1** Department of Prosthodontics, Biomaterials Research Center, School of Dentistry, Shiraz University of Medical Sciences, Shiraz, Fars, Iran, **2** Department of Bacteriology and Virology, Shiraz HIV/AIDS Research Center, Institute of Health, Medical School, Shiraz University of Medical Sciences, Shiraz, Fars, Iran

◉ These authors contributed equally to this work.
‡ These authors also contributed equally to this work.
* derafshi@sums.ac.ir

**Data Availability Statement:** All data are available within the attached files as Supporting information.

## Abstract

Fabricating method may affect the surface properties and biological characteristics of provisional restorations. This study aimed to evaluate the surface roughness, plaque accumulation, and cytotoxicity of provisional restorative materials fabricated by the conventional, digital subtractive and additive methods. Sixty-six bar-shaped specimens (2×4×10 mm) were fabricated by using provisional restorative materials through the conventional, digital subtractive and additive methods (n = 22 per group). Ten specimens of each group were used for surface roughness and plaque accumulation tests, 10 specimens for cytotoxicity assay, and 2 specimens of each group were used for qualitative assessment by scanning electron microscopy. The $R_a$ (roughness average) and $R_z$ (roughness height) values (μm) were measured via profilometer, and visual inspection was performed through scanning electron microscopy. Plaque accumulation of *Streptococcus mutans* and cytotoxicity on human gingival fibroblast-like cells were evaluated. The data were analyzed with one-way ANOVA and Tukey's *post hoc* test (α = 0.05). Surface roughness, biofilm accumulation and cytotoxicity were significantly different among the groups (P<0.05). Surface roughness was significantly higher in the conventional group (P<0.05); however, the two other groups were not significantly different (P>0.05). Significantly higher bacterial attachment was observed in the additive group than the subtractive (P<0.001) and conventional group (P = 0.025); while, the conventional and subtractive groups were statistically similar (P = 0.111). Regarding the cytotoxicity, the additive group had significantly higher cell viability than the subtractive group (P = 0.006); yet, the conventional group was not significantly different from the additive (P = 0.354) and subtractive group (P = 0.101). Surface roughness was the highest in conventionally cured group; but, the additive group had the most plaque accumulation and lowest cytotoxicity.

**Funding:** R.G.: Vice-Chancellery of Research of Shiraz University of Medical Sciences (grant #98-01-03-20246).

**Competing interests:** The authors have declared that no competing interests exist.

## Introduction

Provisional restorations are the inseparable part of fixed prosthodontics and dental implant treatments. They protect the pulpal tissue against thermal, mechanical, physical and bacterial contaminations [1, 2]. A well-made provisional restoration also matters esthetically and keeps the soft tissue healthy until definitive restoration is delivered [3, 4].

Provisional restorations are made with several methods and materials like the conventional chairside materials, which are so commonly used for direct and indirect restorations. But, mixing the powder and liquid to fulfill external surface mold could develop voids and negatively affect the mechanical and surface properties. Some common disadvantages are associated with the conventional method such as polymerization shrinkage, thermal damage of the pulp cells, porous surface, lack of marginal adaptation, water absorption, and color instability [2]. It also highly relies on the technician's skills, and those with several abutments are much time-consuming and difficult to be made through the conventional method [5, 6].

The computer-aided design and computed-aided manufacturing (CAD-CAM) is being increasingly used for making dental restorations. In subtractive manufacturing method, the provisional restorations are milled out of a prefabricated poly methyl methacrylate (PMMA) block, with a high degree of conversion, accuracy, strength, marginal adaptation and color stability [7]. However, there are some flaws like the positive and negative errors in diameter of milling burs that may cause inaccuracies, besides the waste of material, and insufficiency in milling complex shapes [8–11].

In the additive manufacturing system, the product is manufactured by consequtively piling up the powder and liquid type of materials [5, 12]. The ability to print complicated orders besides no waste material are the major superiorities of additive method over subtractive method [3, 8, 10]. Many studies compared the mechanical properties of different provisional materials with respect to their fabrication methods [1, 3, 11, 13]. Yet, limited information is available about the biological behavior of provisional restorations manufactured through digital methods.

Biofilm accumulation on dental martials causes gingival inflammation, denture stomatitis, and secondary caries. *Streptococcus mutans* contribute to the development of caries and that biofilms are more resistant than planktonic microorganisms [14]. Compared with the diffinitive restorations, the provisionals have higher surface roughness and less marginal adaptation, which cuase more biofilm attachment on their surfaces. Rough surfaces increase the initial attachment of bacteria to the provisional restorative materials by protecting them from saliva and masticatory forces [15, 16]. Sufficient polishing can decrease the surface roughness and plaque acuumulation [17–19]. Recent articles compared the surface roughness and biofilm formation between the provisional restorations made through digital and conventional methods [6, 20, 21].

Temporization phase of prosthetic treatment also aims to develop a healthy soft tissue around the margin of the prepared tooth, under pontics and around implant abutments [22]. These restorations help shaping the soft tissue profile around the implant abutments to achieve the esthetic demands [19]. Biocompatible materials are of great importance particularly in complicaed treatment plans with prolonged use of provisional restorations [19, 23].

Limited studeis have compared the biocompatibility of digitally fabricated provisional restorations and conventioal ones [9, 19, 24–27]. The present study aimed to compare the surface and biological charachtristics of provisional restoration fabricated by three differents methods. The null hypothesis believed that the fabrication methods would not affect the surface ruoghness, biofilm accumulation, and cytotoxicity.

## Materials and methods

### Specimen preparation

Sixty-six bar-shaped specimens (10×4×2 mm) were fabricated through one of the conventional, digital subtractive and additive methods (n = 20 per group). Ten specimens of each group were used for testing the surface roughness and plaque accumulation, 10 specimens were used for cytotoxicity assay, and 2 specimens of each group were used for qualitative evaluation by scanning electron microscopy. Bar-shaped specimens were preferred to the disk-shaped ones to reduce the waste material in digital subtractive method and also because it suited all tests in the present study.

In the conventional method, a custom-made silicon rubber mold with bar-shaped holes was fabricated. Then, the powder and liquid of auto-polymerized PMMA acrylic resin (Tempron; GC, Japan) were mixed and packed into the mold in early dough stage. In digital subtractive method, the computer file in STL format was transferred to the milling machine (Ceramill Motion 2; Amann Girrbach AG, Germany) and bar-shaped specimens were milled out of pre-polymerized PMMA blanks (Yamahachi Dental MFG. Co.; Japan). In digital additive method, the specimens were printed by a digital light processing 3D printer (Asiga MAX UV; Austria) with layering thickness of 1 μm by using a resin-based provisional material (Freeprint temp; DETAX GmbH & Co. KG, Germany). All specimens were finished and polished by using 400, 600 and 800 grits of sandpapers (Aewio; China) [28].

### Surface roughness and characterization assays

The mean surface roughness ($R_a$ [μm]) and the arithmetic mean height of the surface profile ($R_z$ [μm]) of 10 specimens of each group were measured with a contact profilometer (Rogosurf 20; TESA, Switzerland) with 0.25 mm cutoff length, 4 mm transverse length, 0.001 μm resolution and 1 mm/s speed of stylus. For each specimen, 3 measurements were made and the mean values were recorded. For qualitative characterization, 2 specimens of each group were gold coated with a sputter coater (S150B; Edwards, UK) and examined at 15 kV by using a scanning electron microscope (SEM, JSM-6335 F; JEOL, Japan) at ×500 and ×1500 magnifications. After measuring the surface roughness values, 10 specimens of each group were sterilized under ultra violet wave (59S UV sterilizer; China) for 30 minutes on both sides before plaque accumulation test [9].

### Plaque accumulation assays

Following a standardized method three colonies of a reference strain of *S. mutans (ATCC 35668)* was cultured overnight (16 hours) in brain heart infusion broth at 37°C in an anaerobic atmosphere after being checked by Gram-staining and catalase activity. The bacterial suspension was adjusted to an optical density of 0.09 at 600 nm. Optical density measurement was based on a previously calculated optical density/bacterial count gradient curve ($10^8$ CFU/mL). Two mL of ultra-filtered tryptone yeast extract broth supplemented with 1% sucrose and 20 μL of adjusted bacterial suspension were pipetted in 24-well plate containing 10 sterile samples for each group for biofilm formation. The formed biofilm on each specimen was washed 3 times daily in 0.9% NaCl to remove the unattached bacteria and then transferred to a new plate with fresh ultra-filtered tryptone yeast extract broth containing 1% sucrose for 24 hours. All plates were incubated at 37°C in an environment of 5 to 10% $CO_2$ in anaerobic condition. After 72 hours, the specimens were washed 3 times in 0.9% NaCl and transferred to microcentrifuge tubes containing 1 mL of 0.9% NaCl. A sonicator at 30 W (Branson SFX150-Y SFX250-Y SFX550-Y, China) detached the microorganisms from the specimens. Then, 100 μL

of the biofilm suspension was 10 fold serially diluted up to $10^6$ and 20 μL of each suspension was incubated in brain heart infusion at 37˚C for 72 hours in anaerobic atmosphere. After the incubation, the colony forming units (CFU) in plates with 30 to 300 typical colonies of *S. mutans* were counted using a Darkfield Quebec Colony Counter (Reichert Technologies, New York, USA) and then reported in CFU/mL [29]. This process was also performed on the uncultured negative control plates to rule out any contamination.

## Cytotoxicity assays

Vials of human gingival fibroblast (HGF1-PI1(NCBI C165)) were provided from the cellular bank of Pasteur Institute of Iran. Cells were cultured in flasks with Dulbecco's modified Eagle medium (DMEM, Biowest, Nuaillé, France) containing 15% fetal bovine serum (Biowest, Nuaillé, France) and 1% glutamine–penicillin–streptomycin (Biowest, Nuaillé, France), incubated at 37˚C, 90% humidity and 5% $CO_2$. Then, $7 \times 10^3$ of cells in 50 μL of culture medium were added in 12-well plates and incubated for 4 hours at 37˚C to be attached on ten specimens of each group. The same cell concentration was cultured on empty 12-well plates as negative control group, and polyurethane was used as the positive control in the MTT assay (Sigma St. Louis, MO, USA). The polyurethane was cut in sections of 4×2 mm and subjected to the MTT assay as of the original samples. The cell viability was about 11% for the positive control and 100% for the negative control. Then, 500 μL of incubation medium was added to each well and incubated for 72 hours, The medium was removed and 400 μL of 3(4,5-dimethylthiazol-2-yl)-2,5-diphenyltetrazoliumbromid with 0.5 mg/mL concentration were added to each well. All plates were incubated for 4 hours, the medium was discarded afterwards and 400 μL of isopropanol was added. The formazan was solubilized on a shaker for 15 minute. Then, 100 μL of solution was added to 90-well plates and optical density of each well were assessed by Elisa meter (STAT FAX 2100, USA) in 570 nm [30]. Cytotoxicity responses were rated via the following formula:

$$\text{Cell viability} = \frac{sample\ mean\ OD}{control\ mean\ OD} \times 100$$

Cell viability above 90% were considered as non-cytotoxic, 60 to 90% as slightly cytotoxic, 30 to 59% as moderately cytotoxic, and those below 30% inferred severe cytotoxicity [31].

SPSS Statistics for Windows software version 17.0 (SPSS Inc, Chicago, IL, USA,) was used for statistical analysis. Kolmogorov-Smirnov test assessed the hypothesis of normal distribution. One-way ANOVA and Tukey's HSD post-hoc test were used to compare the mean cell viability (%) and plaque accumulation among the groups. To compare the surface roughness parameters, Kruskal-Wallis H and Dunn's post-hoc test were used. Type I error rate was considered to be α = 0.05. To compute the effect sizes, partial eta squared ($\eta_p^2$) and eta H squared ($\eta_H^2$) were used for one-way ANOVA F and Kruskal-Wallis H tests, respectively.

## Results

### Surface roughness

Based on the results of Kruskal-Wallis test, the three groups were significantly different in terms of $R_a$ (μm) and $R_z$ (μm) surface roughness parameters (P = 0.004 and P = 0.006, respectively). The conventional group had significantly higher surface roughness than the subtractive ($R_a$ (μm): P = 0.010, $R_z$ (μm): P<0.001) and additive groups ($R_a$: P = 0.004, $R_z$: P = 0.004). The subtractive and additive groups were not statistically different in $R_a$ (P = 0.754) and $R_z$ (P = 0.673) (Table 1, Fig 1). SEM analysis confirmed the rougher surface of the conventional

**Table 1. Mean rank (mean ± standard deviation) of the surface roughness (μm).**

| Surface roughness parameters<br><br>Groups | $R_a$ (μm) | $R_z$ (μm) |
|---|---|---|
| Conventional | 22.00 [a] (1.35 ± 0.71) | 21.70 [a] (12.89 ± 6.8) |
| Subtractive | 8.85 [b] (0.53 ± 0.09) | 9.10 [b] (4.49 ± 1.34) |
| Additive | 15.65 [b] (0.67 ± 0.15) | 15.70 [b] (4.09 ± 2.14) |
| ES* | 0.26 | 0.23 |
| P† | 0.004 | 0.006 |

*: Eta H squared ($\eta_H^2$) effect size.

†: P-value for Kruskal-Wallis H test.

Mean rank values with different letters in superscript were statistically significant (Dunn's post-hoc test).

group compared with the two other groups. Digitally fabricated specimens had more homogenous surfaces (Fig 2).

## Plaque accumulation

The mean values of plaque accumulation (CFU/mL) were converted into logarithmic (log 10) values and analyzed with one-way ANOVA. *S. mutans* accumulation was significantly different among the groups (P<0.001); being significantly higher in the additive group than the conventional (P = 0.025) and subtractive groups (P<0.001). But, the conventional and subtractive groups were statistically similar (P = 0.111) (Table 2, Fig 3).

## Cytotoxicity

ANOVA test revealed that the cell viability (%) was significantly different among the groups (P = 0.007). Additive group had significantly higher cell viability than the subtractive group (P = 0.006). Yet, no significant difference existed between the additive and conventional (P = 0.354) and conventional and subtractive groups (P = 0.101) (Table 2, Fig 4).

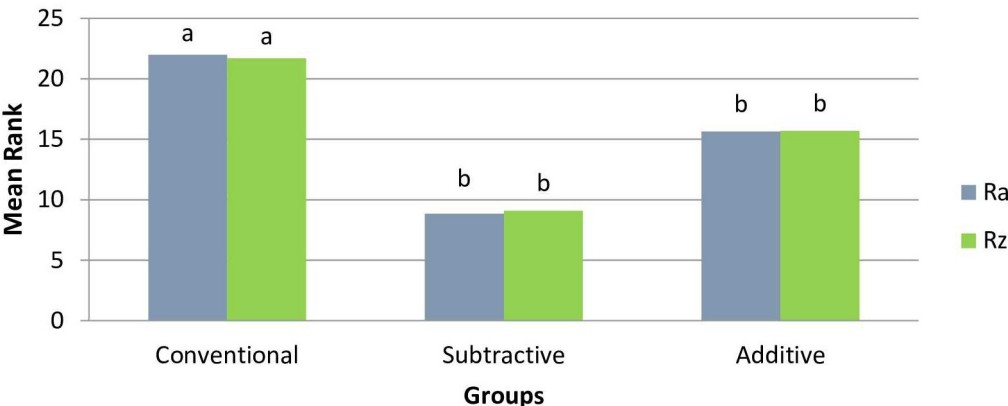

**Fig 1. Mean rank of surface roughness parameters ($R_a$ and $R_z$).**

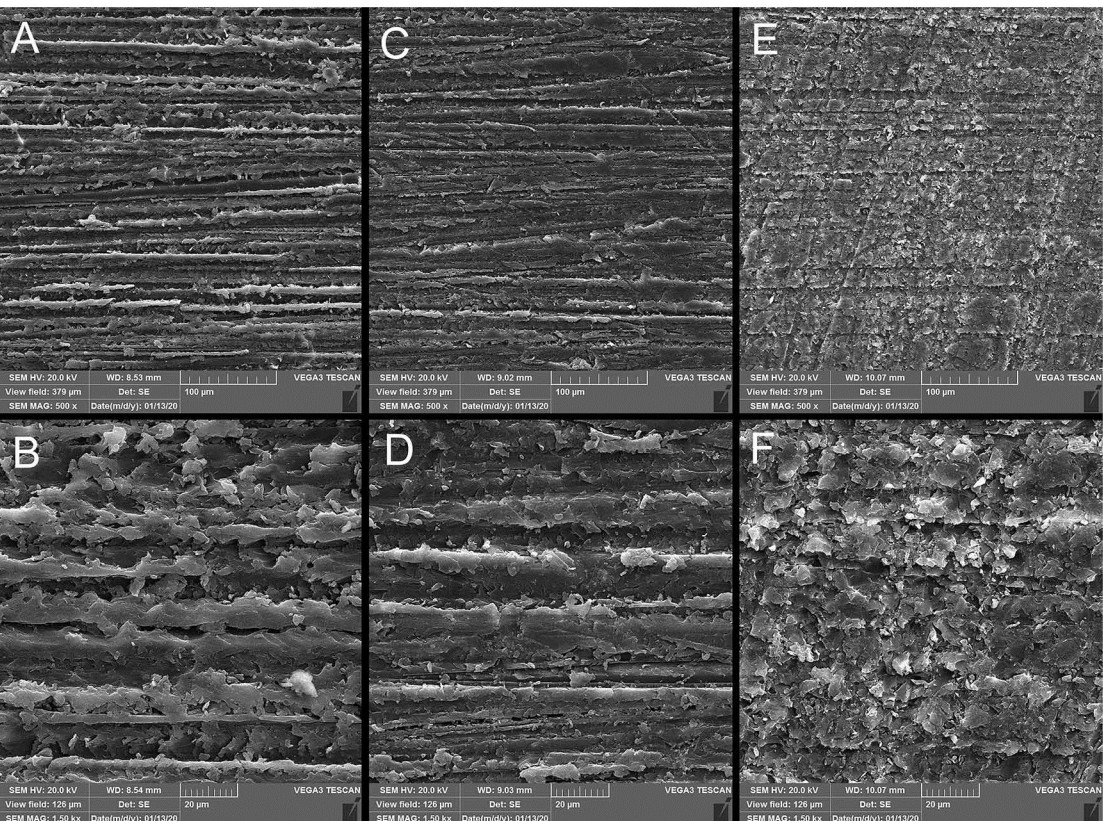

**Fig 2. Scanning electron micrograph analysis (×500, ×1500 magnifications).** A and B: conventional; C and D: subtractive, E and F: additive.

## Discussion

The null hypothesis was wholly rejected as the material property used in each fabrication method significantly affected the surface roughness, plaque accumulation, and cytotoxicity of provisional materials. Surface roughness is one of the most important criteria that affects the biofilm accumulation on dental materials [21, 32]. This study evaluated both $R_a$ and $R_z$

**Table 2. Mean ± standard deviation of plaque accumulation and cell viability.**

| Biological characteristics <br> Groups | Plaque accumulation $10^6$ (CFU/mL) | Cell viability (%) |
|---|---|---|
| Conventional | 7.89 ± 0.28 [b] | 91.18 ± 7.63 [ab] |
| Subtractive | 7.63 ± 0.25 [b] | 81.20 ± 10.82 [a] |
| Additive | 8.24 ± 0.28 [a] | 97.64 ± 6.58 [b] |
| ES* | 0.60 | 0.42 |
| P† | <0.001 | 0.007 |

*: Partial eta squared ($\eta_p^2$) effect size.

†: P-value for one-way ANOVA F test.

Mean values with different letters in superscript were statistically significant (Tukey's post-hoc test).

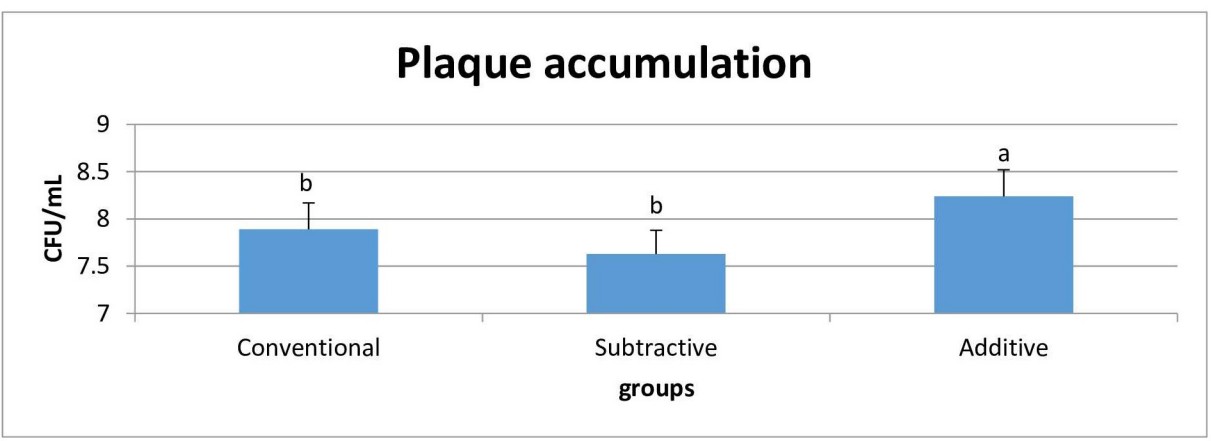

**Fig 3. Mean and standard deviation of plaque accumulation.**

values because in case of equality of the $R_a$ values, less $R_z$ values imply smoother surface [33]. Although all the surface roughness values in this study were higher than the threshold (0.2 μm) that could eliminate the role of surface roughness in plaque adherence [34, 35], they were below the limit for clinical undetectability of roughness (10 μm) [16].

In the present study, conventionally cured PMMA had significantly higher surface roughness than the digitally fabricated groups. Similar findings were reported by some other researchers [16, 19, 20], namely Meshni et al. [21], who reported higher surface roughness in conventionally cured PMMA resins than in modified methyl methacrylate resins and CAD-CAM PMMA blocks. They observed the lowest surface roughness in CAD-CAM PMMA blocks.

Simoneti et al. [6] compared two 3D printed provisional restorative materials manufactured through laser stereolithography and selective laser sintering with conventional PMMA and bis-acrylic resins, and found higher surface roughness in the conventional PMMA than the printed groups. Moreover, the current study showed no significant difference between the surface roughness of subtractive and additive groups in neither $R_a$ and $R_z$ values. Presumably,

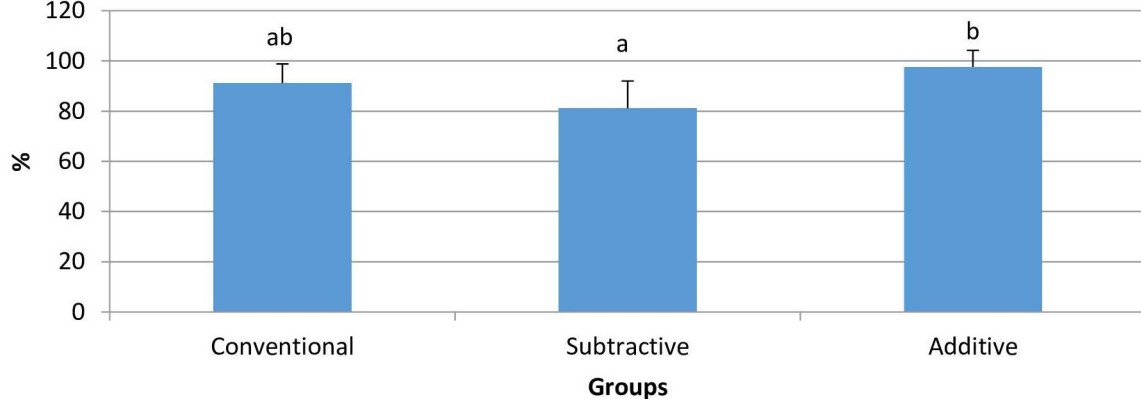

**Fig 4. Mean and standard deviation of cell viability.**

the high surface roughness of the conventional group is due to the air bubbles incorporated through hand mixing of liquid and powder during filling of external mold [2, 16].

Although higher plaque accumulation was expected in the conventional group with higher surface roughness, the 3D printed samples showed the highest attachment of *S. mutans*; whereas, the conventional and subtractive groups showed similarly lower bacterial adhesion. Likewise, Simoneti et al. [6] detected higher *S. mutans* attachment on the 3D printed provisional materials than the conventionally cured acrylic resin. The low bacterial colonization in the subtractive group was similarly reported by another study that compared the bacterial colonization between CAD-CAM PMMA blocks and conventionally cured provisional materials [21].

Özel et al. [36] attributed the low attachment of *S. mutans* on PMMA surfaces to its higher surface energy and the subsequent higher hydrophobicity of PMMA. They also considered the effect of residual methyl methacrylate monomers in PMMA resin on the cell viability of *S. mutans*. Attachment of *S. mutans* was also reported to be lower on PMMA acrylic resins than bis-acrylate provisional resin materials [15].

Although surface roughness is so determining, bacterial attachment also depends on the chemical composition, surface topography and free energy, as well as hydrophobicity [15, 21]. Thus, the heterogeneous composition due to hydrophobic resin matrix and hydrophilic filler particles with different sizes, weights and chemical contents may explain the different tendency of *S. mutans* on bis-acrylate and conventional PMMA resins [37, 38]. Besides, a study reported significantly higher biofilm attachment on materials with more urethane di-methacrylate rather than BisGMA and TEGDMA in their matrix [37].

Polishing can expose some filler particles on the surface of heterogeneous material, and consequently affect plaque accumulation [38]. X-ray photo electron spectroscopy revealed that polishing the resin based materials decreased the carbon and increased the silicon content, which consequently influenced the surface energy of substrate [18]. Supposedly, the chemical composition accounts for the higher attachment of *S. mutans* on 3D printed provisional material in the present study. Pituru et al. [23] asserted that printable provisional materials usually contain monomers based on acrylic esters or filled hybrid materials. However, precise comparison is not possible as manufacturers do not provide adequate information about exact content of matrix and filler particles of these materials [12, 13, 23, 39].

Cytotoxicity assay is essential to evaluate the biocompatibility of dental materials [9]. Cell toxicity of resin materials is referred to leaching out of residual monomers or other eluates inducing genotoxic effects [23]. Elution of residual monomers depends on the chemical composition, degree of conversion, and solvents in in-vivo conditions [19]. To the best of the authors' knowledge, no study has ever compared the cytotoxicity of 3D printed provisional materials with those fabricated through digital subtractive and conventional methods. Since cell viability was the highest in the additive, followed by the conventional and subtractive group, the additive and conventional groups were considered as non-cytotoxic and the subtractive group as slightly cytotoxic [31].

Higher cell viability in the additive group could be due to its chemical composition, including monomers based on acrylic esters or filled hybrid materials that are not clearly disclosed by the manufacturers [23]. Another possible could be the high degree of conversion due to post-polymerization procedure in this fabricating method [30].

Slight cytotoxicity was observed in the subtractive group. A similar study comparing the cytotoxicity of different polymer and ceramic CAD-CAM materials found that the prefabricated PMMA blank (VITA CAD-Tem) showed slight cytotoxicity on gingival fibroblasts after 72 hours [25]. Engler et al. [24] investigated the residual monomer elution from different conventional and CAD-CAM dental polymers. They detected higher monomer elution from the

conventional PMMA materials that stabilized in the first seven days but Teilo- CAD (CAD-CAM PMMA) started higher monomer elution after 48 hours, continuing until the 60th day of aging. Continuous monomer release was observed in low quantities in CAD-CAM polymers.

Apparently, prefabricated PMMA blanks are not completely inert and may differ in the degree of conversion based on the manufacturer companies. In contrast, studies have documented the cytotoxic effect of conventionally cured PMMA on fibroblast cells among provisional materials [40], as well as high cell viability in CAD-CAM pre-polymerized blocks [9, 26, 27].

Another study accounted the chemical composition for the different values of flexural strength of several brand of CAD-CAM PMMA blanks. They reported higher flexural strength in M-PM-disc than in Polident PMMA and Teilo CAD due to the organic modified polymer-network in M-PM structure [41]. It can be concluded that different brands of CAD-CAM PMMA blanks have different chemical compositions and degrees of conversion, which may affect their mechanical and biological properties. It explains the slight cytotoxicity of Yamaha-shi PMMA CAD-CAM blank used in the present study.

It is so important to choose an appropriate fabrication method with the least negative effect on the biocompatibility and surface properties of provisional restorations, particularly for those on long span with prolonged use. Due to the lower surface roughness and acceptable biofilm formation and cytotoxicity of two digital manufacturing techniques than the conventional chairside method and particularly lower chairside time and clinical appointments, these two digital methods may be more suitable for fabrication of provisional restorations, especially for long term applications.

Limitations of the present study included the in-vitro use of only one brand of material for each manufacturing techniques for surface properties and biocompatibility assay of these fabrication methods. Besides, lack of accurate information about the chemical composition of digitally fabricated materials restricted the comparisons. Further in-vivo investigations are needed to compare the surface properties and biocompatibility of these fabrication methods for different brands of provisional restorative materials.

## Conclusions

Based on the present findings, it can be concluded that:

1. Conventionally cured PMMA resin showed significantly higher surface roughness than the subtractive and additive digitally fabricated provisional specimens.

2. *S. mutans* accumulate significantly more on provisional restorations fabricated through digital additive method.

3. The specimens fabricated through the conventional and digital additive methods were not cytotoxic to human gingival fibroblast-like cells; yet, those made by using digital subtractive method were slightly cytotoxic.

## Supporting information

**S1 Table. Data for surface roughness.**
(XLSX)

**S2 Table. Data for plaque accumulation.**
(XLSX)

**S3 Table. Data for cell viability.**
(XLSX)

## Acknowledgments

This article was based on the postgraduate thesis by Dr. Shima Dabiri. Appreciations are also expressed to Dr. Mehrdad Vossoughi and Dr. Seyed Ali Tabei for helping with the statistical analysis and Ms. Farzaneh Rasooli for proofreading, editing, and improving the use of English in this manuscript.

## Author Contributions

**Conceptualization:** Rashin Giti, Shima Dabiri, Mohammad Motamedifar, Reza Derafshi.

**Data curation:** Mohammad Motamedifar, Reza Derafshi.

**Formal analysis:** Rashin Giti, Mohammad Motamedifar, Reza Derafshi.

**Funding acquisition:** Shima Dabiri.

**Investigation:** Shima Dabiri.

**Methodology:** Rashin Giti, Shima Dabiri, Mohammad Motamedifar, Reza Derafshi.

**Project administration:** Rashin Giti, Shima Dabiri.

**Resources:** Rashin Giti, Mohammad Motamedifar.

**Software:** Mohammad Motamedifar.

**Supervision:** Rashin Giti.

**Validation:** Shima Dabiri, Mohammad Motamedifar, Reza Derafshi.

**Visualization:** Rashin Giti, Reza Derafshi.

**Writing – original draft:** Shima Dabiri.

**Writing – review & editing:** Rashin Giti, Shima Dabiri, Mohammad Motamedifar, Reza Derafshi.

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
