## [Decision Letter · Decision Letter 0]

4 Feb 2021

PONE-D-21-00025

Surface roughness, plaque accumulation, and cytotoxicity of provisional restorative materials fabricated by different methods

PLOS ONE

Dear Dr. Derafshi,

Thank you for submitting your manuscript to PLOS ONE. After careful consideration, we feel that it has merit but does not fully meet PLOS ONE’s publication criteria as it currently stands. Therefore, we invite you to submit a revised version of the manuscript that addresses the points raised during the review process.

While the reviewer and editor found the work to be interesting, the manuscript needs a lot of editing and clarification (see reviewer's comments). More importantly, the sample size seems to be too small and not justified. Please consider adding more samples. Also include the power of the study or sample size justification.

We look forward to receiving your revised manuscript.

Kind regards,

Sompop Bencharit, DDS, MS, PhD, FACP

Academic Editor

PLOS ONE

Journal Requirements:

Reviewers' comments:

Reviewer's Responses to Questions

**Comments to the Author**

1. Is the manuscript technically sound, and do the data support the conclusions?

Reviewer #1: Partly

2. Has the statistical analysis been performed appropriately and rigorously? 

Reviewer #1: N/A

3. Have the authors made all data underlying the findings in their manuscript fully available?

Reviewer #1: Yes

4. Is the manuscript presented in an intelligible fashion and written in standard English?

Reviewer #1: No

5. Review Comments to the Author

Reviewer #1: This interesting study aimed to evaluate surface roughness, plaque accumulation and cytotoxicity of provisional restorative materials obtained by the conventional method using a chairside material, or the CAD- CAM (additive and subtractive methods). In general, I suggest the English review of the manuscript by a native speaker. Some grammatical errors must be corrected. The term “interim” should be replaced by “provisional” through the text.

The main fail of this study, in my opinion, is the absence of control groups (positive and negative) for the biological tests. It strongly compromises the study, if it couldn´t be provided.

Abstract:

Sixty bar specimens were fabricated, n=20 for each material? Was it enough? How many different occasions have been performed for the tests?

Introduction:

The Introduction is well-written and refers to a concise and recent bibliography. The authors justified this in vitro study in a clinical perspective. The hypotheses were well elaborated.

Line 42: Please add a reference to the sentence “It has a high degree of conversion, accuracy, strength, marginal adaptation and color stability” regarding the subtractive method.

Line 46: “In digital additive method, a three-dimensional (3D) printer pull up the liquid and powder to design interim restorations”. It’s not mandatory mixing the powder and liquid in the additive method. The materials used for the printing method are usually liquid in a bottle.

On the 5th paragraph, please refer to S. mutans, since it was the evaluated microorganism in the study.

Lines 55-56: “Rough surfaces increase the initial attachment of bacteria by protecting them from saliva and masticatory forces”. Add reference on roughness in provisional restorative materials.

Materials and methods:

1- Is the conventional resin Tempron; GC, Japan auto-polymerized or heat-polymerized?

2- Sixty bar specimens were fabricated, n=20 for each material? Was it enough? Considering that 2 specimens of each group were used for the qualitative evaluation (SEM), only 18 left for the other tests.

3- Please justify the shape of the specimens. Were bar-shaped specimens used for all the tests?

4- Shouldn't the specimens have been standardized as to their roughness surface to perform the biological test? Why didn't the authors do it?

5- Lines 88-89: The roughness parameters “0.25 mm cutoff length, 4 mm transverse length, 0.001 μm resolution and 1 m/s speed of stylus”. Please revise the speed of stylus: 1m/s or 1 mm/s?

6- Were both sides of the specimens sterilized? Does UV sterilization occur before or after the roughness surface test?

7- Using the growth curve of the ATCC35668 strain performed by the authors, what is the time and value of the Optical Density at 600nm corresponding to the exponential phase for the inoculum? What was the CFU count of the inoculum?

8- Why the formation of the acquired film was not carried out before the biofilm accumulation?

9- Line 102: “These procedures were repeated for 3 times”. The experiments for plaque accumulation/ cytotoxicity were performed in triplicate? If so, recalculate the number of specimens.

10- How many different occasions have been taken to compare the mean and standard deviation?

11- Describe the control groups (positive and negative) for plaque accumulation/ cytotoxicity assays.

Results

1- Include the unit (μm) on the description of the roughness results.

2- Include the letters over the bars to display the difference among the groups (Tukey test) in Figures 1, 3 and 4.

3- Lines 130-132 “The mean and standard deviations were calculated and analyzed with one-way ANOVA, Tukey’s post hoc test, and Kruskal-Wallis test (α=0.5)”. Which statistical test was used for each variable? Kruskal-Wallis test performs the comparison among mean ranks, and not means.

Discussion

The discussion is well-written. A concise literature was used to discuss the results.

1 – I suggest that in the description of null hypothesis rejection it will be clear that it was the MATERIAL property used in each manufacturing method that influences the results.

2- Line 168: It has been stated that Ra 0.2 micrometers is the threshold for adherence of microorganisms for denture base materials. I suggest the inclusion of the reference: Zissis AJ, Polyzois GL, Yannikakis SA, Harrison A. Roughness of denture materials: a comparative study. Int J Prosthodont. 2000;13(2):136-140).

3- Please describe the limitations and clinical relevance of the study.

Conclusions:

The conclusions are succinct and display the main findings of the study.

6. PLOS authors have the option to publish the peer review history of their article (what does this mean?). If published, this will include your full peer review and any attached files.

Reviewer #1: No

---

## [Author Response · Author response to Decision Letter 0]

18 Feb 2021

1) The sample size seems to be too small and not justified. Please consider adding more samples. Also include the power of the study or sample size justification.

All effect sizes were calculated for the tests and reported in the tables in the revised version of manuscript. The significant results (p<0.05) and large values for effect size indicated that the sample size was large enough to detect differences among the groups. Partial eta squared (ηp2) and eta H squared (ηH2) were reported for one-way ANOVA F and Kruskal-Wallis H tests, respectively. Although we were not able to compute post power values for nonparametric test, the power values were greater than 80% for one-way ANOVA done for Cell viability (98.2%) and Plaque accumulation (100%) variables.

2) This interesting study aimed to evaluate surface roughness, plaque accumulation and cytotoxicity of provisional restorative materials obtained by the conventional method using a chairside material, or the CAD- CAM (additive and subtractive methods). In general, I suggest the English review of the manuscript by a native speaker. Some grammatical errors must be corrected. The term "interim" should be replaced by "provisional" through the text.The main fail of this study, in my opinion, is the absence of control groups (positive and negative) for the biological tests. It strongly compromises the study, if it couldn´t be provided.

The term "interim" was replaced by "provisional" throughout the text. Also the positive and negative controls for plaque accumulation and cytotoxicity assays were explained in detail in the Materials and Method section.

The manuscript was thoroughly edited by a skilled editor.

3) Sixty bar specimens were fabricated, n=20 for each material? Was it enough? How many different occasions have been performed for the tests?

The number of specimens for each group in each test was precisely described in the abstract and materials and method sections of the revised manuscript: Sixty-six bar-shaped specimens (2×4×10 mm) were fabricated by using provisional restorative materials through the conventional, digital subtractive and additive methods (n=22 per group).10 specimens of each group were used for surface roughness and plaque accumulation tests , 10 specimens were used for cytotoxicity assay and 2 specimens of each group were used for qualitative evaluation by scanning electron microscopy.

All effect sizes were calculated for tests and reported in the tables in the revised version of manuscript. The significant results (p-values<0.05) and large values for effect size indicated that the sample size was sufficiently large to detect differences between the groups. Partial eta squared (ηp2) and eta H squared (ηH2) were reported for one-way ANOVA F and Kruskal-Wallis H tests, respectively. Although we were not able to compute post power values for nonparametric test, the power values were greater than 80% for one-way ANOVA done for Cell viability (98.2%) and Plaque accumulation (100%) variables.

4) Line 42: Please add a reference to the sentence "It has a high degree of conversion, accuracy, strength, marginal adaptation and color stability" regarding the subtractive method.

Relevant reference was cited.

5) Line 46: "In digital additive method, a three-dimensional (3D) printer pull up the liquid and powder to design interim restorations". It's not mandatory mixing the powder and liquid in the additive method. The materials used for the printing method are usually liquid in a bottle.

The statement was edited and citation was added. 

6) On the 5th paragraph, please refer to S. mutans, since it was the evaluated microorganism in the study.

The specific sentence related to S. mutans was added to this paragraph according to the reviewer's precise comment.

7) Lines 55-56: "Rough surfaces increase the initial attachment of bacteria by protecting them from saliva and masticatory forces". Add reference on roughness in provisional restorative materials.

Reference was added.

8) Is the conventional resin Tempron; GC, Japan auto-polymerized or heat-polymerized?

The conventional resin Tempron; GC, Japan was auto-polymerized, which was mentioned in the revised manuscript

9) Sixty bar specimens were fabricated, n=20 for each material? Was it enough? Considering that 2 specimens of each group were used for the qualitative evaluation (SEM), only 18 left for the other tests.

The number of specimens for each group in each test was precisely described in the abstract and materials and method sections of the revised manuscript: Sixty six bar-shaped specimens (2×4×10 mm) were fabricated by using provisional restorative materials through the conventional, digital subtractive and additive methods (n=22 per group).10 specimens of each group were used for surface roughness and plaque accumulation tests , 10 specimens were used for cytotoxicity assay and 2 specimens of each group were used for qualitative evaluation by scanning electron microscopy.

All effect sizes were calculated for tests and reported in the tables in the revised version of manuscript. The significant results (p-values<0.05) and large values for effect size indicated that the sample size was sufficiently large to detect differences between the groups. Partial eta squared (ηp2) and eta H squared (ηH2) were reported for one-way ANOVA F and Kruskal-Wallis H tests, respectively. Although we were not able to compute post power values for nonparametric test, the power values were greater than 80% for one-way ANOVA done for Cell viability (98.2%) and Plaque accumulation (100%) variables.

10) Please justify the shape of the specimens. Were bar-shaped specimens used for all the tests?

The reason of using bar-shaped specimens rather than the disk-shaped ones was related to reducing the waste material during digital subtractive method and also it was appropriate for all tests in the present study: this part was added to the Materials and Method section of revised manuscript.

11) Shouldn't the specimens have been standardized as to their roughness surface to perform the biological test? Why didn't the authors do it?

All the specimens were polished as in the clinical situations, and we wanted to evaluate the surface roughness of different groups after this routine clinical protocol of polishing.

12) Lines 88-89: The roughness parameters "0.25 mm cutoff length, 4 mm transverse length, 0.001 μm resolution and 1 m/s speed of stylus". Please revise the speed of stylus: 1m/s or 1 mm/s?

It was corrected.

13) Were both sides of the specimens sterilized? Does UV sterilization occur before or after the roughness surface test?

Both sides of the specimens were sterilized after surface roughness test and before plaque accumulation test. It was precisely mentioned in the revised manuscript.

14) Using the growth curve of the ATCC35668 strain performed by the authors, what is the time and value of the Optical Density at 600nm corresponding to the exponential phase for the inoculum? What was the CFU count of the inoculum?

These items were added in the text as follow:

Bacteria was cultured overnight (16 hours) in brain heart infusion broth at 37 °C in an anaerobic atmosphere after being checked by Gram-staining and catalase activity. The bacterial suspension was adjusted to an optical density of 0.09 at 600 nm. Optical density measurement was based on a previously calculated optical density/bacterial count gradient curve (108 CFU/mL).

15) Why the formation of the acquired film was not carried out before the biofilm accumulation?

It was done before the biofilm accumulation and for more clarity the text was revised accordingly.

16) Line 102: "These procedures were repeated for 3 times". The experiments for plaque accumulation/ cytotoxicity were performed in triplicate? If so, recalculate the number of specimens.

This was mistakenly mentioned, so it was deleted in the revised manuscript.

17) How many different occasions have been taken to compare the mean and standard deviation?

As mentioned in the revised manuscript, for the ten specimens used for surface roughness and plaque accumulation tests, the plaque accumulation test was done after the surface roughness measurements of the specimens. Also ten other specimens were simultaneously used for the cytotoxicity assay. 

18) describe the control groups (positive and negative) for plaque accumulation/ cytotoxicity assays.

The positive and negative controls for plaque accumulation and cytotoxicity assays were completely explained in the Materials and Method section.

19) Include the unit (μm) on the description of the roughness results.

This unit (μm) was added to the description of the roughness results.

20) Include the letters over the bars to display the difference among the groups (Tukey test) in Figures 1, 3 and 4.

The mentioned letters to display the difference among the groups was added to the bars in Figures 1, 3 and 4.

21) Lines 130-132 "The mean and standard deviations were calculated and analyzed with one-way ANOVA, Tukey's post hoc test, and Kruskal-Wallis test (α=0.5)". Which statistical test was used for each variable? Kruskal-Wallis test performs the comparison among mean ranks, and not means.

One-Way ANOVA and Tukey’s HSD post-hoc test were used to compare mean cell viability (%) and plaque accumulation. To compare surface roughness parameters, Kruskal-Wallis H and Dunn’s post-hoc test were used. The type I error rate was considered to be α=0.05. To compute effect sizes, partial eta squared (ηp2) and eta H squared (ηH2) were used for one-way ANOVA F and Kruskal-Wallis H tests, respectively. This part was precisely added to the Materials and Method section of revised manuscript. Also according to the precise comment of reviewer the mean ranks were reported in Table 1 and Figure 1, which was related to the Kruskal-Wallis test for surface roughness parameters.

22) I suggest that in the description of null hypothesis rejection it will be clear that it was the MATERIAL property used in each manufacturing method that influences the results.

The description of null hypothesis rejection, was revised according to the reviewer’s comment

23) Line 168: It has been stated that Ra 0.2 micrometers is the threshold for adherence of microorganisms for denture base materials. I suggest the inclusion of the reference: Zissis AJ, Polyzois GL, Yannikakis SA, Harrison A. Roughness of denture materials: a comparative study. Int J Prosthodont. 2000;13(2):136-140).

The mentioned reference was added to this section according to the reviewer’s comment.

24) Please describe the limitations and clinical relevance of the study.

The limitations and the clinical relevance of this study were added to the end of Discussion in the revised manuscript.

---

## [Decision Letter · Decision Letter 1]

22 Mar 2021

Surface roughness, plaque accumulation, and cytotoxicity of provisional restorative materials fabricated by different methods

PONE-D-21-00025R1

Dear Dr. Derafshi,

We’re pleased to inform you that your manuscript has been judged scientifically suitable for publication and will be formally accepted for publication once it meets all outstanding technical requirements.

Kind regards,

Sompop Bencharit, DDS, MS, PhD, FACP

Academic Editor

PLOS ONE

Additional Editor Comments (optional):

Thank you for the revision.

Reviewers' comments:

Reviewer's Responses to Questions

**Comments to the Author**

1. If the authors have adequately addressed your comments raised in a previous round of review and you feel that this manuscript is now acceptable for publication, you may indicate that here to bypass the “Comments to the Author” section, enter your conflict of interest statement in the “Confidential to Editor” section, and submit your "Accept" recommendation.

Reviewer #1: All comments have been addressed

2. Is the manuscript technically sound, and do the data support the conclusions?

Reviewer #1: Yes

3. Has the statistical analysis been performed appropriately and rigorously? 

Reviewer #1: Yes

4. Have the authors made all data underlying the findings in their manuscript fully available?

Reviewer #1: Yes

5. Is the manuscript presented in an intelligible fashion and written in standard English?

Reviewer #1: Yes

6. Review Comments to the Author

Reviewer #1: All the questions were answered accordingly. The manuscript was entirely rewritten. The English was reviewd. I suggest for the acceptance of the manuscript.

7. PLOS authors have the option to publish the peer review history of their article (what does this mean?). If published, this will include your full peer review and any attached files.

Reviewer #1: No

---

## [Editor Report · Acceptance letter]

24 Mar 2021

PONE-D-21-00025R1 

Surface roughness, plaque accumulation, and cytotoxicity of provisional restorative materials fabricated by different methods 

Dear Dr. Derafshi:

I'm pleased to inform you that your manuscript has been deemed suitable for publication in PLOS ONE. Congratulations! Your manuscript is now with our production department. 

Kind regards, 

on behalf of

Dr. Sompop Bencharit 

Academic Editor

PLOS ONE